# Entrepreneurial ecosystems in cities: The role of institutions

David Bruce Audretsch[1], Maksim Belitski[2]*, Nataliia Cherkas[3]

**1** Institute of Development Strategies, SPEA Indiana University, Bloomington, Indiana, United States of America, **2** Henley Business School, University of Reading, Whiteknights, Reading, United Kingdom, **3** Institute of Higher Education, Kyiv National Economic University, Kyiv, Ukraine

* m.belitski@reading.ac.uk

## Abstract

Entrepreneurship activity varies significantly across cities. We use the novel data for 1,652 ecosystem actors across sixteen cities in nine developing and transition economies during 2018–2019 to examine the role that institutional context plays in facilitating the productive entrepreneurship and reducing the unproductive entrepreneurship. This study is the first to develop and test a model of multi-dimensional institutional arrangements in cities. It demonstrates that not just that institutions matter in shaping the entrepreneurship ecosystem in cities, but in particular those institutional arrangements enhancing the productive and reducing unproductive entrepreneurship. Our findings suggest that differences between normative, cognitive, and regulatory pillars are associated with variance in both types of entrepreneurship in cities. For the formation of productive and high-growth entrepreneurs, all three pillars of institutional arrangement matter. For unproductive entrepreneurship normative pillar of institutions and the role of civil society matter most. This study has theoretical and practical implications for entrepreneurship ecosystem policy in cities.

## 1. Introduction

In a rush to promote entrepreneurial activity, decent work and economic growth as well as innovation and industry development [1, 2], both policy makers and researchers have embraced the concept of the entrepreneurial ecosystem [3–5]. Policymakers and researchers rush to explain the evidence coming from different parts of the world where some cities thrive and other vanish.

A systematic empirical evidence demonstrates the conducive institutional arrangements [6–9] effect that entrepreneurial ecosystems can have in enhancing entrepreneurship [10–14].

However, what if the entrepreneurial activity emanating from an entrepreneurial ecosystem is not productive [15–17]? In introducing the concept of productive, unproductive and destructive entrepreneurship, in the research of [16] was pointed out that entrepreneurial activity can also detract from economic performance [17, 18]. Despite all the impressive progress made in the literature in fleshing out both what comprises as well as the impact of entrepreneurial ecosystems, the case of Baumol's types of entrepreneurship activity and how best promote productive and avoid unproductive entrepreneurship remains noticeably absent.

**Data Availability Statement:** All relevant information regarding data availability are in the supporting information.

**Funding:** Maksim Belitski (MB) received funding from British Academy Newton Fund No NMGR1180259 and University of Reading the

RETF Research Output prize. The project focused on data collection on entrepreneurial ecosystems in cities The funding does not affect the decision to publish, or preparation of the manuscript.

**Competing interests:** The authors have declared that no competing interests exist.

One of the reasons for this omission in the entrepreneurship ecosystem literature may be that the focus has generally been on the context of the developed economies [4, 14, 19, 20], where Baumol's unproductive entrepreneurship is less prevalent. In fact, a compelling set of studies has found that the institutional context associated with developing and transition countries with changing societal and economic institutions [6, 21], with their heterogeneity of rules, norms and culture [22] is more conducive to unproductive entrepreneurship [18, 23–25].

This paper aim is to examine how various institutional arrangements (pillars) influence both the productive and unproductive entrepreneurial activity in entrepreneurial ecosystems in cities. To date, scholarly progress in this area has been limited mainly by a measurement challenge [11, 26–28]—the measures in use fail to capture the multi-facet and heterogeneous nature of the institutional context phenomena for entrepreneurship. Our work contributes to the institutional and entrepreneurship ecosystems literature by introducing and examining a novel, multi-dimensional institutional context at city-level in developing and transition economies, capturing variation that, we argue, affects both the productive and unproductive of entrepreneurial activity in a city. We draw from institutional theory [29, 30] to create the three dimensions of institutional arrangement–regulatory, cognitive, and normative accomplish this task. We also introduce an emerging role of civil society to facilitate productive and reduce unproductive entrepreneurship in ecosystems.

This study also contributes to regional entrepreneurship and urban studies literature by demonstrating that not just that institutions matter in shaping the entrepreneurial ecosystem quality but in particular those institutional arrangements that enhance government support to entrepreneurship, promoting entrepreneurial culture, sustainability, civil society, and business education.

Our results find a statistically significant association between our measures of cognitive, regulatory and normative institutional arrangements and the productive entrepreneurial activity. In contrast to our predictions, the association between the civil society within the normative pillar and the quality of entrepreneurship was partly supported. Instead, we found that civil society and civil awareness [31] is important to reduce unproductive entrepreneurship activity.

In the next section, we set the foundations of our theoretical argument and present our hypotheses. We then describe an institutional arrangement in developing and transition economies. Section 4 describes the sample, variables, and method. Section 5 reports the main results, while section 6 discusses them. Section 7 concludes with a summary of the main findings, contributions, limitations, and future research.

## 2. Conceptual framework

### 2.1. Institutional theory and entrepreneurship ecosystems

The concept of institutions is multifaceted. It embraces different topics across a wide range of social science fields ranging from an economic perspective [32] and a sociological perspective of institutions [29, 33]. Williamson's hierarchical approach builds on [32] legacy of institutions examining how the complexity of cultural, political, and legal frameworks influence economic development. According to [32], institutional arrangements define incentives that guide individual and firm rational choices. [32] distinguishes between formal "rules of the game" comprising laws and regulations and informal or unwritten codes comprising social arrangements that either impede or foster business activity.

Following this line of argument, entrepreneurs, like any other individuals and organizations, will be influenced by the institutional context in which they operate, and their strategies

will respectively reflect the opportunities and limitations defined by this context [15–17, 34–36]. [15, 16] argues that institutional arrangements that define a prevailing system of payoffs will influence individual efforts between different types of entrepreneurial activity, whether this is productive, unproductive, or destructive. A set of framework conditions based on excessive regulation of business activities, high level of corruption, and poor protection of property rights may produce undesired economic outcomes stimulating the development of shadow economy or leading to a misallocation of resources and capturing transfer of existing wealth that in Baumol's terminology is defined as unproductive entrepreneurship [17]. To facilitate economic growth, policy-makers are urged to develop institutions that will reward entrepreneurs for engaging in the creation of wealth through growth-oriented and productive entrepreneurial activity and penalize entrepreneurs who choose unproductive entrepreneurship [15, 16, 26, 37].

A conducive entrepreneurial ecosystem which is characterized by property rights protection, an efficient system of contract enforcement, and limited government's ability to transfer wealth through taxation and regulation will foster individual activity to launch productive entrepreneurship [13, 22, 38].

Drawing on the works of [26] and [29] on the role of the institutional context for entrepreneurial activity and more recent works of [10, 11, 36] on the governance of entrepreneurial ecosystems that stimulate productive and high-growth entrepreneurship in regions and cities, one can distinguish the definitive role of institutions in the outcomes of the entrepreneurial ecosystem.

The role of institutions is exacerbated by two distinguished viewpoints on the evolution of entrepreneurial ecosystems—the top-down approach and the bottom-up approach in their management [10]. While the bottom-up approach assumes that ecosystems evolve like natural ecosystems, the mechanisms that facilitate this evolution are deeply embedded into the regulatory and cognitive framework of a place [24] and norms, culture, and attitudes towards entrepreneurship and the role of entrepreneurs in society. While this approach contrasts the classical theory in economics governed by an "invisible hand", the institutional theory provides important answers on how culture, rules, norms, and behaviors shared by a specific group of individuals as informal institutions as well as formal regulation [39] spur a variety of entrepreneurship activities. While there is a growing interest in how EEs are governed and evolve in developed economies, only a few studies have been conducted exploring the cases of emerging and developing economies where institutions lead EE evolvement [40].

Three institutional pillars were identified based on the works of [29] and [30], underpinnings entrepreneurship activity in various geographical contexts and ecosystems. These are regulative, cognitive, and normative institutions that provide incentives for entrepreneurial behavior. Entrepreneurship scholars have drawn on an institutional framework to explain the role of changes in the institutional context at a country-level [17, 18] and regional level [14, 23, 24] to explain what drives differences in entrepreneurial activity across various entrepreneurial and innovation ecosystems [9, 41–44].

Our study draws on the role of institutional context by validating the role of the three pillars that either facilitate or impede productive entrepreneurial activity in the ecosystem.

## 2.2. Three pillars of institutions and entrepreneurship activity

Following [29], we apply the following three pillars of institutions to predict the changes in individuals' behavior towards productive or unproductive entrepreneurship in the ecosystem. First is the regulatory pillar comprising regulations, laws, and other support for entrepreneurship that defines the "rules of the game" and legal boundaries. Second is the normative pillar

underpinning social values, norms, and beliefs which govern individual and organization behavior. Third "cognitive" pillar constitutes the "shared logics of action" among individuals and organizations, which they use to interpret available information and formulating their expectations about the outcome of their behavior and select market strategies. Altogether three pillars change the behavior of interconnected in the ecosystem economic actors (e.g., entrepreneurs, policymakers, investors, banks, etc.) [5, 11] towards productive or unproductive entrepreneurship [15, 28, 45].

The regulatory pillar facilitates and hinders entrepreneurship activity by shaping the level of risk involved in the formation and start of a business [26] as formal interactions with economic agents are influenced by the rules established by the government [46]. The regulatory pillar may change the breadth and the depth of resources that the government made available for entrepreneurs and lead to unproductive entrepreneurship activity. Productive entrepreneurial activity is the outcome of opportunities created by greater government support to entrepreneurship and formal institutional networks made accessible and transparent for all economic agents in the entrepreneurship ecosystem. Empirical work that validates the role of context on entrepreneurship finds that high-growth and productive entrepreneurship is higher in the economies with wider government support to entrepreneurship (e.g. reduction of taxes, lower start-up costs) [47, 48] and well-designed formal networks [13, 49, 50].

Several government policies influence the supply side of entrepreneurship, such as the Small Business Innovation Research (SBIR) [51, 52]. While the impact of government programs on the performance of firms has been mixed, most of the studies have been limited to analyzing how the changes in government support may shape productive entrepreneurship activity and reduce unproductive entrepreneurship. For entrepreneurs in the developing country context, government programs can serve as an alternative source of support as economic agents lack financial and labor resources compared to entrepreneurs operating in developed countries.

The role of government support and formalization of entrepreneurs' networks in developing countries may increase new business entry rates, and in particular firm-startups that are most ambitious and growth-oriented [22]. The regulatory pillar is therefore represented by a combination of regulation and direct government interventions related to an increase in the supply of side to entrepreneurs and increase their formal collaboration and connectivity channels [53]. The effect of the regulatory pillar aims to increase productive entrepreneurship in developing countries and at the level of cities and other small unites with city policymakers competing with each other for high business growth rate and innovation. Local government programs can shield entrepreneurs from the adverse consequences of local corrupt authorities and weak local institutions, reducing unproductive entrepreneurship activity [17]. We hypothesize:

**Hypothesis 1: In an entrepreneurship ecosystem, the regulatory pillar of institutions facilitates productive entrepreneurship and reduces unproductive entrepreneurship.**

The cognitive institutional pillar constitutes the set of conditions of the reality and nature, references, schemas, and scripts specific to a socio-cultural context of a certain city, which helps to create the cognitive frameworks through which entrepreneurs preserve information, synthesize, analyze and produce knowledge. The cognitive ability of entrepreneur's changes with the changes in how information is created, shared, and interpreted [29]. Cognitive frameworks are adopted through social interactions [35]. For example, [30] define the cognitive dimension as comprised of the knowledge and skills possessed by the people and which is used to establish and operate new firms. Authors argue that knowledge becomes institutionalized,

which means it is shared between individuals on a certain territory. Individuals may be incentivized to perform a high-growth entrepreneurial activity, should they be able to perceive the readiness and availability of knowledge and skills to identify new opportunities and exploit them [54].

Findings of [30] and [49] suggest that ability to start businesses may be particularly prevalent among individuals with entrepreneurship and business education. Latent entrepreneurs may want to move from their latent stage and start a business should they share the business opportunity's perception [55]. Education may affect individuals by providing them with a sense of autonomy and the skills to innovate, resulting in high-growth and productive entrepreneurship [56]. On the contrary, the absence of business education individually or in the entrepreneurial team may either impede entrepreneurship activity or result in unproductive and latent entrepreneurship [55]. Educational capital does not only explain productive entrepreneurial activity [37], but also results in the entrepreneurial ability to distinguish between profitable and non-profitable market opportunities [54].

The variance of entrepreneurial cognitions across different countries is described in [57], while [36] illustrate using cultural diversity, knowledge, and entrepreneurial dynamics that cognitive proximity of a region is positively associated with starting a business in that region. While there are cross-country differences related to entrepreneurial cognition, opportunity-seeking behavior and culture of entrepreneurship also vary between cities in the same country and between cities cross-country [58], differences in perceptions about the importance of entrepreneurs across different cities in the United States (e.g., stories of entrepreneurs, philanthropists, venture capitals, scale-ups) explain the distribution of entrepreneurship capital and growth of entrepreneurship [59]. Important in entrepreneurial cognition is recognizing opportunities in your city to become an entrepreneur, such as personal experience or connection to someone who is an entrepreneur and who started a business in the previous years [26, 60]. Cross-city differences in culture, perceptions of entrepreneurial activity, and the place given to an entrepreneur are likely to influence entrepreneurial activity via changes in the perceptions of entrepreneurial opportunity and the positivity of entrepreneurial outcomes [54]. Prior empirical research suggests that the local culture of entrepreneurship [36, 61] may influence perceived entrepreneurial opportunities and the intensity of entrepreneurial effort towards high-growth and ambitious entrepreneurship [22]. In some instances, the effect may be as strong as the effect of business regulation. This effect is intensified by the presence or informal of entrepreneurial networks and their capacity to promote and sustain interactions between entrepreneurs as platforms through which individual entrepreneurs engage in the sense-making activity [62] and share the market experience. Social capital developed within the community and in particular via informal entrepreneurial networks is seen as an important determinant of recognition and exploitation of entrepreneurial opportunities [26, 63]. The extant literature that examines entrepreneurial informal networks has focused on the importance of role models find that networks and entrepreneurial actors as nodes in such frameworks. Role models facilitate ambitious entrepreneurship, and the effect is stronger when networks are locally concentrated with a greater flow of tacit knowledge [64]. Role models localized in cities inspire other entrepreneurs from the network community to embark on innovative and high -growth entrepreneurship. Transparency of such networks and co-location of economic actors further embeds individuals into the cognitive framework and prevents entrepreneurs from launching unproductive and destructive activities [15, 16, 65]. Accordingly, we hypothesize:

**Hypothesis 2: In an entrepreneurship ecosystem, the cognitive pillar of institutions increases productive entrepreneurship and reduces unproductive entrepreneurship.**

The normative pillar shapes social behavior through a system of societal values, beliefs, and norms. They are typically viewed as the standards of behavior established, for example, by close social networks (family and friends), professional associations, and business groups, which underlie organizational goals and objectives [35, 66]. Values and beliefs of social groups influence entrepreneurial intentions to the extent of communicating a message to individual entrepreneurs of the relative desirability of their activity [67]. Such beliefs may be embedded in a wider setting of social references shaped by national culture [26].

A diverse and significant range of economic actors constitutes civil society, such as the 'third sector', social enterprises and public agencies, co-operatives and associations [31, 68–70]. Civil society plays an important role in shaping societal values, beliefs, and norms and is represented by a community of citizens [71].

It has been argued that cultural values, including the degree to which people prefer to work as individuals rather than in groups, willing to accept inequality and tolerate risk such favors assertiveness, competition, and success, spur innovative solutions and entrepreneurial culture [72]. Differences between regions and cities that are more individualistically-oriented may foster entrepreneurship activity as it creates stronger independent action and a positive perception of uncertainty and risk [37].

As described by [29], collective and individual values determine the desired goals or standards, while norms detail the means for pursuing these goals [26, 73]. The context of culture in cities changes the norms of human behavior. Norms of human behavior, it's individualistic vs. collective orientation, the level of trust to institutions [74] and other values shared socially, embedded and transmitted by individuals are established based on the acceptance and support a certain behavior [75]. Places that promote innovation and growth-oriented entrepreneurship will confer high status on entrepreneurs as compared to other places which values are towards conservatism, incremental change, or places that face significant crises or economic transitions and where large businesses and the role of government is paramount.

Incorporating these insights in an entrepreneurial ecosystem setting, norms, and values that favor innovation behavior and entrepreneurialism, as well as the degree of social responsibility of individuals, their ability to participate in decision-making for society will influence the economic and social desirability of growth-oriented entrepreneurship while deferring unproductive and destructive entrepreneurship.

The norms that view civil society as co-producers of innovation and growth has been driven by 'third-way' politics and the broader recognition of civil society's capability to spur innovation and entrepreneurial leadership to champion economic growth [71].

Normative institutional pillar comprises the civil society institutions that occupy the place between the State and the private sector [31], and within this place, entrepreneurship can embed social innovations and high-growth entrepreneurship for a greater good [76, 77].

The attitudes and expectations of individuals about their role in society changes the role that entrepreneurs see themselves as agents of welfare and change and further contribute to society for a good cause [67]. If the expectations and beliefs about entrepreneurship contribute to the resolution of societal issues and challenges via innovation and economic growth, they will embrace productive entrepreneurial intentions resulting in an increase in growth-oriented entrepreneurship. On the contrary, entrepreneurs who act in the context of poor civil society and low status on entrepreneurs as agents of change will be more likely to choose an unproductive entrepreneurial activity that neither harms nor helps society. We hypothesize:

**Hypothesis 3: In an entrepreneurship ecosystem, the normative pillar of institutions increases productive entrepreneurship and reduces unproductive entrepreneurship.**

## 3. Institutional context and entrepreneurship ecosystems in transition economies

Institutional contexts change the dynamics of entrepreneurial activity [23, 24, 78] and entrepreneurial perception of opportunities [54].

An excessive protective regulatory pillar in countries in transition or the absence of such pillar, on average, negatively affects entrepreneurship and decreases firm entry [48, 79, 80]. There is a significant issue of trust between entrepreneurship and the regulatory pillar of institutions. For example, in the case of corporate or property tax, if entrepreneurs do not believe that they are not going to receive the benefit from paying the taxes, then they would be less inclined to pay the taxes [81] moving into shadow economy and unproductive entrepreneurship [15, 16].

Developing countries and countries in transition are likely to have a low regulatory pillar of institutions, which reduces the overall effectiveness of the entrepreneurship ecosystem and results in an increase in unproductive entrepreneurship [16, 22, 25, 82]. There are several reasons for this a) collapse or destruction of formerly existing institutions due to gaining independence, war, regime change, or other factors [83]; b) inefficiency of the regulatory framework due to high level of corruption, severe political and/or economic shocks, particularly in the period of transition from one form of governance to the other; c) related to a normative pillar of institutions, such as the absence of strong government traditions, lack of civil society development and limited political and economic freedom [31]; high corruption which also leads to unproductive entrepreneurship activity [18].

Entrepreneurship research has also demonstrated that cognitive and normative pillars of institutions may have a destructive impact on entrepreneurship activity as they create a "grey zone" that is a perfect medium for corruption, nepotism and may destroy or prevent the development of the system of fair goods distribution within the ecosystems [32, 84].

Both arguments have enough evidence confirming them. However, it depends very much on the individual characteristics of specific entrepreneurship ecosystems and the complementarity between various institutional pillars [18, 85].

## 4. Materials and methods

### 4.1. Sample

The institutional theory has proven particularly useful in examining the differences in entrepreneurial activity in transition and developing countries [65, 66, 86] and cities [87].

Our approach suggests that cities are the most appropriate spatial units for this analysis with institutional pillars that are spatially bounded [9, 14, 88]. Thus, our data collection strategy was to limit the study to certain administrative units, like cities [89]. Core-cities provide a more fine-grained level of analysis compared to larger regions, where aggregate additional populations and areas skew the values in an unknown direction.

The authors have thoroughly reviewed the data. Unique features of the survey include sampling for representativeness at the level of city in each country (at least 2 cities in each country), except Bosnia and Turkey; ecosystem size (at least 8 types of ecosystem stakeholders should be present in each ecosystem–professors at university, non-for-profits, government, entrepreneurs, technopark or incubator manager, venture investor, representative of a bank or trust, multinational company C-level manager), and the number of stakeholders in each group (at least 8 adults). Countries were selected building upon the societal clusters proposed by the Global Leadership and Organizational Behavior Effectiveness research program (GLOBE) [90] that groups countries based on cultural dimensions, similar institutions, and economic conditions.

We test our hypotheses by collecting the individual level of economic agents from sixteen cities and nine countries of East Europe (Poland and Ukraine), Caucasus (Georgia), Central Asia (Kazakhstan), southeast Europe (Romania, Bulgaria, and Turkey), and Balkan countries (Croatia and Bosnia and Herzegovina). They are rather peculiar countries, but the representative transition and developing economies. While some countries have advanced in their transition and institutional reforms and joined the European Union (Romania, Poland, Bulgaria, Croatia), other countries continue their institutional reforms. In addition, regional differences are significant in these countries, which provides a diverse sample of sixteen cities, including the largest city of a region–Istanbul.

The process continued by constructing representative cross-country cross-city samples of randomly selected entrepreneurial ecosystem stakeholders in 2018. We started by collecting emails and telephone information (where available) for the 1,943 individuals via the web-pages of the organizations where they work by script with the help of the Phython program using the keywords related to our ecosystem stakeholders (e.g. policy-maker, entrepreneur, lawyer, loan advisor–for banks, journalist, etc.). The records could generally be found by typing their full name, organization. The ensuing email accounts were active. We hired nine research fellows out of Assistant and Associate professors in residence in entrepreneurship to collect the data at Business Schools and management departments across nine representative universities in city capitals. The positions lasted 9 months and were full-time fixed term. Lists of entrepreneurial stakeholders identified by the program were passed over to Research fellows to follow up and contact shortlisted ecosystem stakeholders. Out of the 1,943 individuals identified and emailed, 1652 (85%) responded. Other respondents (15%) either did not respond or refused to complete the surveys or being interviewed. Variables, descriptions, data sources, and descriptive statistics for the study variables are summarized in Table 1 and correlations in Table 2. The data for our survey items was collected through different survey techniques, both online surveys and telephone interviews, during December 2018—January 2020. to avoid common method bias [58]. The characteristics which describe the three pillars of local institutions and entrepreneurship activity building on [26, 27, 29] as the "anchor" for our questionnaire and data collection activities.

Our other sources were the Times Higher Education of the Global University Ranking to identify the number of business schools and management departments in each city as well as environmental air quality data from the IQAir Earth data [91] for environmental awareness and industrial agglomeration in cities.

Our sample contains 51% of observations from capital cities and 49% from regional capitals.

It is important to note that the entrepreneurship ecosystem may overpass core-city boundaries. The 'total population' indicator provides the number of people living within the city but does not include surrounding communities outside of the core city. Therefore, a question may arise whether the surrounding agglomeration zone potentially affects the entrepreneurship ecosystem within larger urban areas.

The distribution of observations is consistent across cities in our sample: Kyiv, Ukraine (7.26%), Lviv, Ukraine (5.81), Wroclaw, Poland (6.17%), Warsaw, Poland (6.17%), Batumi, Georgia (3.63%), Tbilisi, Georgia (7.81%), Astana (6.17%) and Almaty in Kazakhstan (6.36%), Cluj, Romania (6.96%), Bucharest, Romania (7.20%), Istanbul, Turkey (5.45%), Sarajevo (Bosnia and Herzegovina (6.23%), Zagreb (6.96%) and Osijek, Croatia (6.3%), Sofia (5.93%) and Plovdiv, Bulgaria (5.51%). Almost 95% of respondents have a university degree and above.

Our four major groups of stakeholders are entrepreneurs (35.1%), university professors (8.1% of a sample), policymakers (7.4% of a sample), as well as respondents of multiple affiliations (31.9% of a sample). Other stakeholders include investors, a representative from the

**Table 1. Descriptive statistics.**

| Variables | Description of variables | Mean | St. Dev. | Min. | Max. |
|---|---|---|---|---|---|
| Productive entrepreneurship | Do you agree with the statement: There is a strong focus on growth-oriented and productive entrepreneurship activity in my region (city) (1—very weak, 7—very strong) | 4.70 | 1.50 | 1.00 | 7.00 |
| Unproductive entrepreneurship | Do you agree with the statement: There is a unproductive entrepreneurship in my city (economic activity in formal and informal cooperation with local (national) government to access resources in a privileged way compared to other entrepreneurs whose access could be limited or restricted) (1- very weak, 7—very strong) | 4.48 | 1.55 | 1.00 | 7.00 |
| **Regulatory institutional pillar** | | | | | |
| Formal networks | Do you agree with the statement: There is a sufficient formal network to support entrepreneurship EE in my region (city) (1- very weak, 7—very strong) (government grants, collaboration within Triple-Helix partnerships; incubators and accelerators for entrepreneurship, public-private partnerships, etc.) | 3.82 | 1.39 | 1.00 | 7.00 |
| Government support | Do you agree with the statement: There is a sufficient number of government entrepreneurship support programs in my region (city) (1- very weak, 7—very strong) | 3.80 | 1.49 | 1.00 | 7.00 |
| **Cognitive institutional pillar** | | | | | |
| Business schools | Number of Business schools or Management schools and faculties that teach entrepreneurship, strategy, management, and strategic skills with at least one national or international accreditation (AACSB, EPAS, EQUIS, etc.). [96] | 10.93 | 4.54 | 1.00 | 17.00 |
| Culture | Do you agree with the statement: There is a strong entrepreneurship culture and orientation in my region (city) and I personally know entrepreneur who started a business in the previous years (1- very weak, 7—very strong) | 4.16 | 1.61 | 1.00 | 7.00 |
| Informal networks | Do you agree with the statement: There is a sufficient informal network to entrepreneurship in my region (city) (1- very weak, 7—very strong) (knowing angel investors, informal business meetings, business clubs, entrepreneur's families, friends, colleagues and relations with other actors) | 4.39 | 1.51 | 1.00 | 7.00 |
| **Normative institutional pillar** | | | | | |
| Media support | Do you agree with the statement: There is a high status of entrepreneur in my region (city) as well as a sufficient support of independent mass media to entrepreneurship in my region (city) (1- very weak, 7—very strong) | 3.85 | 1.55 | 1.00 | 7.00 |
| Venture capital | Do you agree with the statement: There is a sufficient private equity capital (business angels, venture capital, crowdfunding) in my region (city) to support entrepreneurship (1- very weak, 7—very strong) | 3.48 | 1.55 | 1.00 | 7.00 |
| Environmental awareness | Share or adult residents of a city who are registered with IQAir earth data [91] to monitor and report air quality in total population | 7.83 | 11.18 | 1.13 | 45.63 |
| Sustainability | Do you agree with the statement: There is a strong awareness for sustainability in my city (healthy lifestyle, veganism, energy efficiency, sustainability, corporate social responsibility) in my city (region) (1- very weak, 7—very strong) | 3.66 | 1.51 | 1.00 | 7.00 |
| Civil society | Number of nationally and internationally recognized non-for-profit organizations in my city (region) focused on changing human behavior related to inequality, democracy, civil rights, health and environmental protection, labor market regulation, home abuse) | 17.48 | 3.19 | 7.00 | 20.00 |
| **Control variables** | | | | | |
| Entrepreneur | Area of activity (entrepreneur = 1, otherwise = 0) | 0.35 | 0.48 | 0.00 | 1.00 |
| Professor | Area of activity (professor = 1, otherwise = 0) | 0.09 | 0.28 | 0.00 | 1.00 |
| Multiple | Multiple occupations (entrepreneur, professor, policymaker, investor, director/manager in a multinational company, manager of TTO, manager in techno park (accelerator); lawyer, other) (multiple = 1, otherwise = 0) | 0.34 | 0.47 | 0.00 | 1.00 |
| Gender | Gender (male = 1, female = 0) | 0.56 | 0.50 | 0.00 | 1.00 |
| University degree | Have you got a university degree or higher? (1—yes; 0—no) | 0.84 | 0.36 | 0.00 | 1.00 |
| Age range | Age group (less than 29 years old = 1; 30–39 = 2; 40–49 = 3; 50–59 = 4; 60–69 = 5; more than 70 = 6) | 2.29 | 1.11 | 1.00 | 6.00 |
| Capital city | Capital city = 1, otherwise = 0. | 0.54 | 0.50 | 0.00 | 1.00 |
| Debt capital | Do you agree with the statement: There is a sufficient debt capital (bank and other debt capital providers, financial associations, peer-to-peer lending, business-to-business lending, invoice factoring, etc.) in my region (city) to support entrepreneurship (1- very weak, 7—very strong) | 4.46 | 1.68 | 1.00 | 7.00 |
| Population | Population in logs, World Bank database | 13.75 | 1.10 | 11.59 | 16.56 |
| Air pollution | Rank of city air pollution: one—least polluted and 300 most polluted city in the world. Source: IQAir Earth data [91] | 81.78 | 39.37 | 6.00 | 158.00 |

Source: Authors' elaboration using entrepreneurship ecosystem collected data and [91, 96].

**Table 2. Correlation matrix.**

| | 1 | 2 | 3 | 4 | 5 | 6 | 7 | 8 | 9 | 10 | 11 | 12 | 13 | 14 | 15 | 16 | 17 | 18 | 19 | 20 | 21 |
|---|---|---|---|---|---|---|---|---|---|---|---|---|---|---|---|---|---|---|---|---|---|
| 1.Productive entrepreneurship | 1 | | | | | | | | | | | | | | | | | | | | |
| 2.Unproductive entrepreneurship | -0.16* | 1 | | | | | | | | | | | | | | | | | | | |
| 3.Formal networks | 0.41* | 0.01 | 1 | | | | | | | | | | | | | | | | | | |
| 4.Government support | 0.43* | -0.08* | 0.48* | 1 | | | | | | | | | | | | | | | | | |
| 5.Business schools | 0.06* | -0.16* | 0.05* | 0.10* | 1 | | | | | | | | | | | | | | | | |
| 6.Culture | 0.52* | -0.01 | 0.48* | 0.35* | 0.06* | 1 | | | | | | | | | | | | | | | |
| 7.Informal networks | 0.36* | 0.21* | 0.49* | 0.32* | -0.05* | 0.38* | 1 | | | | | | | | | | | | | | |
| 8.Media support | 0.36* | 0.01 | 0.47* | 0.37* | 0.07* | 0.45* | 0.36* | 1 | | | | | | | | | | | | | |
| 9.Venture capital | 0.41* | 0.03 | 0.48* | 0.47* | 0.04 | 0.39* | 0.35* | 0.42* | 1 | | | | | | | | | | | | |
| 10.Environmental awareness | -0.01 | -0.06* | 0.14* | 0.08* | 0.05* | 0.10* | -0.02 | -0.02 | 0.11* | 1 | | | | | | | | | | | |
| 11.Sustainability | 0.40* | -0.03 | 0.43* | 0.33* | 0.10* | 0.51* | 0.27* | 0.45* | 0.44* | 0.01 | 1 | | | | | | | | | | |
| 12.Civil society | -0.09* | -0.04 | 0.03 | -0.03 | -0.22* | -0.08* | -0.18* | 0.09 | 0.04 | 0.07* | 0.03 | 1 | | | | | | | | | |
| 13.Entrepreneur | 0.05* | 0.02 | -0.11* | -0.12* | -0.11* | -0.04 | 0.06* | -0.07* | -0.09* | -0.10* | -0.09* | -0.15* | 1 | | | | | | | | |
| 14.Professor | 0.01 | -0.01 | 0.09 | 0.02 | -0.01 | 0.07 | -0.02 | 0.02 | 0.07 | -0.02 | 0.09 | 0.03 | -0.22* | 1 | | | | | | | |
| 15.Multiple | -0.08* | 0.03 | 0.03 | 0.06* | 0.03 | -0.02 | -0.06* | 0.02 | 0.04 | 0.03 | 0.05* | 0.05* | -0.52* | -0.22* | 1 | | | | | | |
| 16.Gender | -0.02 | 0.02 | 0.02 | -0.03 | 0.02 | -0.02 | -0.04 | -0.01 | -0.07* | 0.03 | -0.06* | 0.05* | 0.05* | -0.05* | -0.06* | 1 | | | | | |
| 17.University degree | 0.24* | 0.07 | 0.01 | 0.07* | -0.03 | 0.12* | 0.03 | -0.01 | 0.03 | 0.10* | 0.02 | -0.08* | -0.02 | 0.08* | -0.08* | -0.09* | 1 | | | | |
| 18.Age range | -0.06 | 0.08* | -0.04* | -0.01 | -0.07* | -0.08* | -0.14* | 0.09 | -0.05* | 0.05* | -0.03 | 0.08* | -0.06* | 0.19* | -0.09* | 0.07* | 0.15* | 1 | | | |
| 19.Capital city | 0.08 | 0.05* | 0.02 | 0.08* | -0.07 | -0.02 | -0.05* | 0.04 | 0.14* | 0.38* | -0.01* | 0.23* | -0.11* | 0.05* | 0.10* | -0.01 | 0.18* | -0.03 | 1 | | |
| 20.Debt capital | 0.41* | 0.07* | 0.42* | 0.32* | -0.06* | 0.47* | 0.46* | 0.34* | 0.35* | -0.01 | 0.27* | -0.14* | 0.01 | 0.08 | -0.01 | -0.02 | 0.12* | -0.08* | 0.01 | 1 | |
| 21.Population | 0.01 | -0.04 | -0.02 | 0.02 | 0.28* | -0.04 | 0.08* | 0.08* | 0.07* | -0.22* | -0.04 | -0.13* | 0.02 | 0.05* | -0.08* | 0.01 | -0.28* | -0.16* | 0.13* | -0.08* | 1 |
| 22.Air pollution | -0.16* | -0.09* | 0.01 | -0.06* | -0.05* | -0.03 | 0.07* | -0.12* | -0.02 | 0.34* | -0.21* | -0.07* | 0.06* | -0.07* | -0.04 | 0.01 | -0.10* | -0.20* | 0.13* | 0.06* | 0.11* |

Level of statistical significance is * 0.05%. Source: Authors' elaboration using entrepreneurship ecosystem collected data and [91, 96].

chamber of commerce, managers in multinational firms, technology transfer office (TTO) managers, managers in techno park, journalists, managers in business incubators, and lawyers.

Considering the few missing observations, researchers often use averaged indicators to predict the role of institutions in an entrepreneurial activity, which is incorrect as it may produce different results, and causality could not be claimed. This is not the approach we followed, as we excluded all missing data. We follow Sobel (2008) [17], who used cross-sectional estimation for the role of institutions to entrepreneurship in the US states.

## 4.2. Dependent variables

Our dependent variables include measures of productive and unproductive entrepreneurial activity in the ecosystem. We study productive entrepreneurship activity (EE quality) with the following survey question "There is a strong focus on growth-oriented and productive entrepreneurship activity in my region (city)" measured on the Likert scale from 1 –very weak to 7 –very strong [11]. The average value of EE quality is 4.70 and a standard deviation of 1.50. This measure has been used in [16] as well as [17] as well as more recent study on the role of institutions for entrepreneurial quality cross-country [18].

We use the question in a survey which measures the degree of unproductive entrepreneurship in a city based on the studies of [18]. Unproductive entrepreneurship represents the unethical behavior of firms and necessity-driven entrepreneurship activity, which is particularly relevant for transition and developing countries [92]. We define Unproductive entrepreneurship with the following survey question "There is an economic activity of entrepreneurs via formal and informal cooperation with the local (national) government to access resources in a privileged way compared to other entrepreneurs whose access to resources could be limited or restricted measured on the Likert scale from 1 –very weak to 7 –very strong [17].

## 4.3. Independent variables

Table 2 lists the dependent and independent variables used in this study. Independent variables measure the effectiveness of the various components in an ecosystem [93]. Our independent variables are divided into three pillars.

Regulatory pillar. We used two variables to measure the regulatory pillar of city-level institutional arrangements. We used formal networks as the respondent's perception about their efficiency to support entrepreneurship (networks between ecosystem stakeholders such as universities, incubators, accelerators, Chamber of commerce, government grants, Triple-Helix) [5, 94]. In unpacking the role of networks, much attention is paid to the relational elements between multiple actors. We used the availability of government support to entrepreneurship as a second variable (e.g., SBIR program for the US, other Public-private partnerships, etc) [52, 95].

Cognitive pillar. Three variables capture the perception of perceived business opportunities and the skills necessary for starting a business within the adult population in a city. We obtained the number of Business schools or Management schools and faculties that teach entrepreneurship, strategy, management, and strategic skills with at least one national or international accreditation (AACSB, EPAS, EQUIS, etc.) from the World University Rankings 2020 at Times Higher Education data [96].

It is important for -entrepreneurial adults to see promising opportunities to start a business in a city as we asked respondents: "There is a strong entrepreneurship culture and orientation in my region (city) and I personally know an entrepreneur who started a business in the previous years" [26]. Finally, we used informal networks in a survey "There is a sufficient informal

network to entrepreneurship in my city" scaled between one and seven as a proxy for the degree of personal involvement and connection to other EE stakeholders and to entrepreneurs who started a business. Entrepreneurial culture and informal networks provide a city-level reflection of the factors driving entrepreneurial activity, such as the perceived feasibility, viability, and desirability of entrepreneurship [67].

Normative pillar. This pillar consists of two dimensions. The first dimension relates to two variables from the survey to capture the influence of culture, norms in institutional arrangements affecting the entrepreneurial environment in an ecosystem. Our first variable measured the status of entrepreneurship in a city by a survey question "There is a high status of an entrepreneur in my region (city) as well as a sufficient support of independent mass media to entrepreneurship in my region (city) (1- very weak, 7—very strong)". This variable also measures perceived media attention paid to entrepreneurship activity. Our second variable measures the availability of venture capital that indicates the relative level of capital markets support for innovative, risky projects with the survey question: "There is a sufficient private equity capital (business angels, venture capital, crowdfunding) in my region (city)" scaled between 1- very weak and 7—very strong.

The second dimension measures the influence of civil society and sustainable orientation in a city that may change the culture and norms of institutional arrangements and change the behavior and objectives of entrepreneurs in a city. The role that civil society plays in sustainability, human rights, environmental, institutional developments vary between cities in the same country and between countries. We use environmental awareness as a share of residents in a city who are registered with IQAir earth data to monitor and report air quality in total population and express their civic position by such participation [31]. Second, we include the number of nationally and internationally recognized not-for-profit organizations in a city aiming at changing human behavior [76, 77]. Finally, we measure the degree of sustainability orientation in a city [97].

### 4.4. Control variables

We have included several control variables. We use the respondent's occupation as a set of binary variables, gender, human capital (university degree or above), age range [98]. We control for cities agglomeration effects [99] as a binary variable if a city is a capital-city, zero otherwise. Capital cities are known to generate more entrepreneurship, agglomeration effects, and in the region of study are important centers of economic development and growth. We control for Air pollution using the IQAir Earth data as a proxy for industrial agglomeration in cities. Access to the debt for entrepreneurs is used as a proxy for financial resources availability in a city and as a form of financing for entrepreneurship in addition to equity and other venture capital.

### 4.5. Model

To test our hypotheses, we use ordinary least square (OLS) estimation with controls for city and country-specific effects. We follow [100], who considered the regression model to capture the effects within the cross-sectional data given by (1). Inclusion of city and country fixed effects allows to control for other city and country fixed effects that are unobserved. Standard errors are robust for heteroscedasticity in all specifications. As a robustness check, we cluster apply the sensitivity analysis, which includes introducing each institutional pillar once at a time and then jointly control for three pillars together in the final estimation. This approach enables to differentiate between various institutional effects and observe potential complementarity and substitutability of pillars as a form of institutional arrangement. The following

model was estimated:

$$y_i = f(\beta x_{i,} \theta z_{i,} a_i, \mu_i) \quad i = 1, \ldots, N; \tag{1}$$

where $y_i$ is productive (unproductive) entrepreneurship in city $i$. $\beta$ and $\theta$ are parameters to be estimated, $x_{it}$ is a vector of independent explanatory variables in city $i$ related to testing our H1-H3; $z_{it}$ is a vector of control variables such as individual characteristics of respondents in city $i$; $a_{it}$ is a vector of country and city-level fixed effects.

To address the concern of multicollinearity, we used variance inflation factor (VIF) in all models with VIF<5.

## 5. Results

Table 3 presents the results of estimation (1) using sensitivity analysis by introducing each institutional pillar one at a time (specifications 1–4, and 6–9, Table 3). Model 1 illustrates the relationship between the variety of institutional pillars and productive entrepreneurship (spec. 1–5, Table 3) and unproductive entrepreneurship (spec. 6–10, Table 3). We start our sensitivity analysis by introducing a regulatory pillar supporting H1. An increase in the perception of the efficiency of government support to entrepreneurship by 1 unit increases productive entrepreneurship by 0.254 (spec. 1 Table 3) and reduces unproductive entrepreneurship by 0.156 (spec. 6 Table 3). While the formal network results are positive and significant for productive entrepreneurship, once we control for another institutional arrangement, such as formal networks variable is no longer significant (spec. 5, Table 3). Once we control for other institutional arrangements, the effect of the regulatory pillar is reduced but remains positive on its impact on productive entrepreneurship ($\beta$ = 0.174, p<0.01). The effect of the regulatory pillar on unproductive entrepreneurship is negative ($\beta$ = -0.171, p<0.01).

We find that the cognitive institutional pillar in a city is positively associated with the productive entrepreneurial activity, with positive effects of entrepreneurial culture in a city ($\beta$ = 0.250, p<0.01) and informal networks ($\beta$ = 0.057, p<0.05) (spec. 5, Table 3). Our hypothesis 2 is supported. Interestingly that the presence of the business school is positively associated with productive entrepreneurship (spec. 1, Table 3), however, the effect is not significant while controlling for other institutional arrangements (spec. 5, Table 3). We find that the cognitive pillar in a city has a mixed effect on unproductive entrepreneurial activity. An increase in the number of business schools in a city is negatively associated with unproductive entrepreneurship ($\beta$ = -0.039, p<0.01), however the informal networks may facilitate unproductive entrepreneurship ($\beta$ = 0.316, p<0.01). The results remain robust when we control for other institutional arrangements. H2 in this instance, is partly supported.

Hypothesis 3 suggesting that the normative pillar of institutions increases productive entrepreneurship is supported as private equity capital ($\beta$ = 0.105, p<0.01) and sustainability awareness ($\beta$ = 0.066, p<0.01) (spec. 5 Table 3). Civil society and the number of non-for-profits are unlikely to change the rate of productive entrepreneurship as the coefficients are is not statistically significant (spec 5, Table 3). Interestingly, that the effect of environmental awareness society is initially positive ($\beta$ = 0.009, p<0.01) (spec. 4 Table 3), however when controlled for other factors the results are no longer significant.

Finally, the normative pillar is negatively associated with the unproductive entrepreneurship supporting H3 as the coefficients of environmental awareness ($\beta$ = -0.008, p<0.01), sustainable orientation ($\beta$ = -0.061, p<0.05) and civil society ($\beta$ = -0.041, p<0.01) are negative and significant. However, in contrast to our hypotheses and the previous studies, we do not find support for our hypothesized influence of other components of the normative institutional arrangements on unproductive entrepreneurship such support of independent mass

**Table 3. Regression analysis of institutional arrangement and entrepreneurship activity in cities.**

| Model | Model 1 –Productive entrepreneurship | | | | | Model 2– Unproductive entrepreneurship | | | | |
|---|---|---|---|---|---|---|---|---|---|---|
| Specifications | (1) | (2) | (3) | (4) | (5) | (6) | (7) | (8) | (9) | (10) |
| **Regulatory institutional pillar** | | | | | | | | | | |
| Formal networks | 0.219*** | | | | 0.044 | 0.049 | | | | -0.027 |
| | (0.03) | | | | (0.03) | (0.04) | | | | (0.04) |
| Government support | 0.254*** | | | | 0.174*** | -0.156*** | | | | -0.171*** |
| | (0.02) | | | | (0.02) | (0.03) | | | | (0.03) |
| **Cognitive institutional pillar** | | | | | | | | | | |
| Business schools | | 0.012* | | | 0.001 | | -0.044*** | | | -0.039*** |
| | | (0.01) | | | (0.01) | | (0.01) | | | (0.01) |
| Culture | | 0.348*** | | | 0.250*** | | -0.097*** | | | -0.035 |
| | | (0.02) | | | (0.02) | | (0.03) | | | (0.03) |
| Informal networks | | 0.140*** | | | 0.057** | | 0.287*** | | | 0.316*** |
| | | (0.02) | | | (0.03) | | (0.03) | | | (0.03) |
| **Normative institutional pillar** | | | | | | | | | | |
| Media support | | | 0.152*** | | 0.005 | | | -0.032 | | -0.011 |
| | | | (0.02) | | (0.02) | | | (0.03) | | (0.03) |
| Venture capital | | | 0.258*** | | 0.105*** | | | 0.007 | | 0.055* |
| | | | (0.02) | | (0.02) | | | (0.03) | | (0.03) |
| Environmental awareness | | | | 0.009*** | -0.001 | | | | -0.013*** | -0.008** |
| | | | | (0.00) | (0.00) | | | | (0.00) | (0.00) |
| Sustainability | | | | 0.283*** | 0.066*** | | | | -0.067** | -0.061** |
| | | | | (0.02) | (0.02) | | | | (0.03) | (0.03) |
| Civil society | | | | -0.009 | -0.006 | | | | -0.043*** | -0.041*** |
| | | | | (0.01) | (0.01) | | | | (0.01) | (0.01) |
| **Other controls** | | | | | | | | | | |
| Gender | -0.012 | 0.001 | 0.022 | 0.008 | 0.029 | -0.021 | 0.024 | -0.004 | 0.003 | 0.049 |
| | (0.06) | (0.06) | (0.06) | (0.06) | (0.06) | (0.07) | (0.07) | (0.08) | (0.07) | (0.07) |
| University degree | 0.926*** | 0.747*** | 0.921*** | 0.919*** | 0.801*** | -0.201* | -0.171 | -0.230** | -0.340*** | -0.268** |
| | (0.09) | (0.08) | (0.09) | (0.09) | (0.08) | (0.11) | (0.11) | (0.12) | (0.12) | (0.12) |
| Age range | -0.014 | 0.033 | -0.005 | -0.010 | 0.020 | 0.113*** | 0.120*** | 0.111*** | 0.121*** | 0.132*** |
| | (0.03) | (0.03) | (0.03) | (0.03) | (0.03) | (0.04) | (0.04) | (0.04) | (0.04) | (0.04) |
| Stakeholder type controls | Yes | Yes | Yes | Yes | Yes | Yes | Yes | Yes | Yes | Yes |
| City and country controls | Yes | Yes | Yes | Yes | Yes | Yes | Yes | Yes | Yes | Yes |
| Capital city | -0.142** | 0.013 | -0.184*** | -0.115 | -0.087 | 0.294*** | 0.288*** | 0.275*** | 0.471*** | 0.475*** |
| | (0.06) | (0.06) | (0.06) | (0.08) | (0.07) | (0.08) | (0.08) | (0.08) | (0.09) | (0.09) |
| Debt capital | 0.212*** | 0.177*** | 0.231*** | 0.294*** | 0.123*** | 0.109*** | -0.005 | 0.090*** | 0.084*** | 0.006 |
| | (0.02) | (0.02) | (0.02) | (0.02) | (0.02) | (0.03) | (0.03) | (0.03) | (0.03) | (0.03) |
| Population | 0.168*** | 0.128*** | 0.121*** | 0.190*** | 0.129*** | -0.041 | -0.037 | -0.043 | -0.124*** | -0.104** |
| | (0.03) | (0.03) | (0.03) | (0.03) | (0.03) | (0.04) | (0.04) | (0.04) | (0.04) | (0.04) |
| Air pollution | -0.006*** | -0.006*** | -0.005*** | -0.005*** | -0.005*** | -0.004*** | -0.004*** | -0.004*** | -0.003*** | -0.005*** |
| | (0.00) | (0.00) | (0.00) | (0.00) | (0.00) | (0.00) | (0.00) | (0.00) | (0.00) | (0.00) |
| Constant | -0.582 | -0.309 | 0.264 | -0.442 | -0.632 | 4.998*** | 4.680*** | 4.801*** | 6.923*** | 6.757*** |
| | (0.46) | (0.46) | (0.46) | (0.58) | (0.52) | (0.61) | (0.59) | (0.61) | (0.76) | (0.72) |
| Number of obs. | 1652 | 1652 | 1652 | 1652 | 1652 | 1652 | 1652 | 1652 | 1652 | 1652 |
| R2 | .381 | .406 | .360 | .336 | .465 | .046 | .108 | .031 | .047 | .142 |
| RMSE | 1.188 | 1.166 | 1.208 | 1.232 | 1.106 | 1.518 | 1.462 | 1.532 | 1.510 | 1.447 |
| F stat | 97.83 | 93.74 | 88.12 | 74.26 | 86.74 | 6.37 | 15.73 | 3.91 | 5.84 | 12.84 |

(*Continued*)

**Table 3.** (Continued)

| Model | Model 1 –Productive entrepreneurship | | | | | Model 2– Unproductive entrepreneurship | | | | |
|---|---|---|---|---|---|---|---|---|---|---|
| loglikelihood | -2688.07 | -2654.24 | -2706.84 | -2755.39 | -2525.39 | -3069.81 | -3013.79 | -3075.88 | -3078.89 | -2944.34 |

*0.01

**0.05

***0.001 significance level.

Source: Authors' elaboration using entrepreneurship ecosystem collected data and [91, 96].

media to entrepreneurship (spec. 10, Table 3). the coefficients of equity capital is marginally significant at 10% and positive, which may demonstrate that equity capital in developing economy may aim to minimize the investment risk by investing in entrepreneurs, supported by corrupt authorities [78].

## 6. Analysis

A striking contrast occurs when we compare the productive and unproductive entrepreneurial activity results in models 1 and 2. Broadly speaking, the relationships between the institutional pillars are reversed when the dependent variable changes from productive to unproductive entrepreneurial activity. As detailed results demonstrated government support to entrepreneurship, Table 3 changes entrepreneurs' behavior with the focus on growth-oriented and productive entrepreneurship. This finding supports prior research on cross-country effects of government support on net productivity score [18] as well as prior research [17]. The effect is being stronger in developing countries due to the supply-side effect of government. At the same time, high-quality government programs can filter unproductive entrepreneurship and attract productive entrepreneurs by creating a system of incentives [17] that are likely to facilitate innovation and the growth aspirations of individuals, increasing the quality of entrepreneurship. Formal networks with entrepreneurs were found a less efficient conduit to unproductive entrepreneurship; however, could facilitate productive entrepreneurship if other institutions are absent or insufficiently developed.

The cognitive institutional arrangements may not be associated with the type of entrepreneurial activity [26] at a country level, however, we find the effect at a city-level. Our results show a strong positive association between informal networks, entrepreneurial culture, on the one hand and productive entrepreneurship on the other hand. Business education that increases entrepreneurial cognition is also likely to support productive and high-growth firms supporting prior research of [30] and [87] for cities. This demonstrates the importance of business education in increasing entrepreneurs' ability in the ecosystem in cities to exploit market opportunities [54].

Curiously, we find informal networks increase the unproductive entrepreneurship supporting prior research on developing economies and the role of corruption in the ecosystem [35, 65, 66].

The relationship between our normative pillar of institutions and the type of entrepreneurial activity, suggesting an intriguing possibility that even if entrepreneurship is a socially acceptable choice and is given high status, pursuing either productive or unproductive and innovation-oriented entrepreneurship is not, supporting prior contradictive research of [26]. Our results for our civil society dimensions contribute to the extant literature on the role of civil society's capability to spur innovation and entrepreneurial leadership to champion economic growth [71]. We provide an empirical test of prior research on the role of civil society

[76, 77] and sustainability of an ecosystem [97], with our finding support the increasing role of sustainable orientation in the productive entrepreneurship. As civic society occupies a place between the State and the private sector, civil society is a powerful conduit to misuse public resources and reduce unproductive entrepreneurship, supporting [31]. In addition to civil society, an increase in sustainability and social awareness is an important factor in directing entrepreneurial action outside of unproductive goals.

Our results are similar to the findings [101] that VCs' positive effect on productive entrepreneurship, but contrasts [18] on the equity venture capital increases both productive and unproductive entrepreneurship. The reason for this contrast is the way [18] measure productive entrepreneurship. They use net entrepreneurship score as a difference between two, while we view both types of entrepreneurship separately, and we were able to find the effects of institutional pillars that affect both types of entrepreneurial activity. Prior research demonstrated that both developed and developing countries use venture capital to increase productive entrepreneurship [102, 103], while we also find that VC supports risk-taking behavior of any type of entrepreneurship activity and may not exclusively target growth-oriented entrepreneurship.

Similar to the availability of private equity capital, informal networks, particularly in EEs with weak formal institutions, may have a side effect of "favorizm" to entrepreneurs with a higher number of connections and their intensity [40, 104]. We found that informal networks in EE positively affect both types of entrepreneurship activity. For example, giving priority to stakeholders in informal networks may affect the quality of the entrepreneurial project, as not always the most competitive can be chosen to the market that is exceptionally relevant for government spending. There may be an informal bias that decreases the efficiency of market mechanisms.

## 7. Summary

Regional policymakers and scholars are in a rush for setting up institutional arrangements for entrepreneurship ecosystem which are conducive for productive entrepreneurship and reduce unproductive entrepreneurship [20, 36, 105]. EEs of higher quality have normative, cognitive, and regulatory pillars of institutions working together as a conduit to productive entrepreneurship, spur innovation and reduce opportunistic behavior of corporates, entrepreneurs and policy makers. Previous conceptualizations of EEs introduced by [3, 10, 49], among others, were focused on the factors characterizing effective EEs in order to identify the most important elements that help entrepreneurs to grow and scale up.

In our study, we draw the attention of policymakers and scholars to measure the role that institutional context plays for two types of entrepreneurship activity between 16 cities in nine developing countries. The introduction of several civil society indicators such as environmental awareness, number of non-for-profits, and sustainability have been advanced as potential solutions to further unpack the role of culture and norms on entrepreneurship in cities.

This study advances a multidimensional measure of a city-level institutional environment and investigates its relationship with the productive and unproductive entrepreneurial activity in the ecosystem in developing and transition economies. It includes three distinct institutional pillars linked to the entrepreneurial activity in a city entrepreneurial ecosystem contributing to the extant literature on institutional context and entrepreneurship [41, 42, 106, 107]. This study engages with the novel measures of normative, cognitive, and regulatory institutional arrangements resulting in further knowledge about local (sub-national institutions). Our results shed new light on the variance between various institutional pillars in their ability to foster or reduce productive and growth-oriented entrepreneurship.

If regional policymakers aim to increase the productive entrepreneurship in their cities, our findings suggest that their emphasis should be on establishing supportive regulative institutional arrangements and government programs, increase informal networks and promote entrepreneurial culture, increase the sustainability orientation of entrepreneurs and the role of civil society. Interestingly, by increasing the supply of venture capital, for example, by liberalizing private equity investment and crowdfunding, we may find it creates a conducive environment for risk-taking, affecting both productive and unproductive entrepreneurship. We find that an environment with regulative institutions reduces unproductive entrepreneurship, along with arrangements of non-for-profits and increasing business educations at business schools and management departments. Our multidimensional measure of city-level institutions reveals a more nuanced relationship between those institutional pillars and the type of entrepreneurship [22, 26, 36, 108]. Policy measures designed to facilitate productive entrepreneurship in the ecosystem would be well served to focus on the normative and cognitive pillars as well as call for government support of entrepreneurs [52, 95]. Our results hint that entrepreneurial ecosystems in cities in transition and less-developed countries may, in fact, benefit more from changes in institutional arrangements that entrepreneurs in developed economies [15, 18, 46, 65].

In contrast to previous studies using country-level data [26, 36] and regional data [23, 24, 109], our results show a more complicated relationship between economic growth and entrepreneurship in developing countries than previously thought. Entrepreneurship might not be always desirable economic activity in the ecosystem and across different institutional contexts [41], or perhaps in some contexts, institutions can cancel each other out or be so complex that they create a "red tape." Our work compliments entrepreneurship ecosystem studies and institutional theory that suggest that the bottom-up approach to entrepreneurial ecosystems is the only right one. On the contrary, we demonstrate that active government support [52, 102] as well as facilitating formal and informal networks [62], entrepreneurial culture, sustainability orientation, and the role of civil society in the economy may might facilitate growth-oriented entrepreneurial activity in ecosystems.

There are several limitations of this study that should be addressed by future research. First, our findings are limited to sixteen cities and nine countries across heterogeneous institutional systems. The cross-sectional nature of our data, while novel, limits our analysis to cross-sectional estimation. Future research could use panel data or big data analysis to develop different clusters of the multilevel institutional framework and systemic conditions to explain that different combinations of institutional arrangements may be able to secure a similar outcome. More data is needed to make the analysis more robust and allow the use of additional analytical techniques (e.g. cluster, semantic and so on).

Second, future work may explore this in more detail using panel data analysis to identify the multidimensional measure of city-level institutions to further unpack their effect on the productive and unproductive entrepreneurial activity. Further research may focus on theorizing the role of civil society institutions that provide the cultural and institutional foundation for future economic growth and entrepreneurial orientation towards productive goals. A well-functioning national cultural and societal framework [37] should be further investigated as moderators of regulatory and cognitive institutional arrangements for a higher rate of entrepreneurship.

## Supporting information

**S1 File. Data availability statement.**
(DOCX)

**S2 File. Dataset ecosystems in cities.**
(DTA)

**S3 File. Dofile for ecosystems in cities data.**
(DO)

## Author Contributions

**Conceptualization:** David Bruce Audretsch.

**Data curation:** Maksim Belitski.

**Formal analysis:** Maksim Belitski, Nataliia Cherkas.

**Funding acquisition:** Maksim Belitski.

**Investigation:** David Bruce Audretsch.

**Methodology:** Maksim Belitski.

**Resources:** Maksim Belitski, Nataliia Cherkas.

**Supervision:** David Bruce Audretsch.

**Validation:** Maksim Belitski.

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
