## [Decision Letter · Decision Letter 0]

28 Oct 2020

PONE-D-20-27940

Entrepreneurial Ecosystems in Cities: The Role of Networks and Institutional Quality

PLOS ONE

Dear Dr. Belitski,

Thank you for submitting your manuscript to PLOS ONE. After careful consideration, we feel that it has merit but does not fully meet PLOS ONE’s publication criteria as it currently stands. Therefore, we invite you to submit a revised version of the manuscript that addresses the points raised during the review process.

Both reviewers suggest substantial revision. The manuscript seems rushed and includes many mistakes. All parts should also be edited/expanded for clarity (see the two reviewers' suggestions). One reviewer asks for more data to cover the role of institutions at multiple scales. If this is not possible, please include more details about this shortcoming in the discussion and edit the paper to reflect the working scale.

We look forward to receiving your revised manuscript.

Kind regards,

Laurentiu Rozylowicz, Ph.D.

Academic Editor

PLOS ONE

Journal Requirements:

Reviewers' comments:

Reviewer's Responses to Questions

**Comments to the Author**

1. Is the manuscript technically sound, and do the data support the conclusions?

Reviewer #1: Partly

Reviewer #2: Yes

2. Has the statistical analysis been performed appropriately and rigorously? 

Reviewer #1: Yes

Reviewer #2: Yes

3. Have the authors made all data underlying the findings in their manuscript fully available?

Reviewer #1: No

Reviewer #2: Yes

4. Is the manuscript presented in an intelligible fashion and written in standard English?

Reviewer #1: Yes

Reviewer #2: Yes

5. Review Comments to the Author

Reviewer #1: This manuscript has the potential to contribute to research on regional entrepreneurship and urban studies. The argument is generally clear and supported by data. The methods are robust. The paper is sufficiently grounded in relevant literature. My recommendations are as follows:

Major recommendation:

Further investigate the role of institutions at multiple scales. Much regional studies/urban studies/economic geography literature emphasizes the importance of analyzing not only national institutions, but also regional and local. Yet local and regional institutions are analyzed rather indirectly here. The authors consider the GEDI index (scaled at the national level) and a survey question about corruption (the question asks whether local 'political entrepreneurs' engage in corrupt practices with national governments). Since this paper is focused on urban entrepreneurship, it seems odd that the paper does not consider the effect of urban policy or urban institutions.

Minor recommendations:

1) Clarify the research question. Perhaps I am misreading, but the research question does not seem relevant to the subsequent hypotheses ('what if the entrepreneurial activity emanating from an entrepreneurial ecosystem is not productive?' p2). The question is about negative factors that inhibit entrepreneurship (e.g. that paragraph references Baumol's 'destructive entrepreneurship' concept). But the hypotheses are about the dynamics of entrepreneurial ecosystems in developing countries (informal networks and institutional quality).

2) Provide more details on the survey sampling method. Why and how were the respondents selected? What makes them representative authorities on their respective cities? And why is the response rate so high (90%)?

Reviewer #2: Overall, the paper is dealing with an important issue the role of networking and connectivity in relation to institutional quality in EE. The main focus of the paper is EE in the developing countries. In these respects, the paper provides novel contribution to the EE literature. The authors use a unique dataset to test their hypothesis. While the paper deserves publishing in Plos One. However, I am suggesting some changes before publishing.

First, it seems that this paper is not the final version that the authors wanted to submit. There are some unfinished sentences and double words. In addition, while there are some references in the result section to hypothesis 3, but I cannot find hypothesis 3, only hypothesis 1 and 2.

The concept section is the weakest part of the paper and should be rewritten by centering on the core concept. Right now it is rather a reading list about the importance of networks and connectivity in the EE. It seems, that the authors follow the Stam (2015) model but it is not explicit. Later they refer to the systemic conditions in the Stam model as it is equal to the network. However, networking is only one element in the Stam concept. I think it needs more clarification. First, what are the elements of the EE and how networking is connected to these elements. The development issue should come after the core model is clear.

Regarding the hypothesis I would add another hypothesis about the effect of culture also.

Regarding Baumol you should refer not only to destructive but also to unproductive entrepreneurship, that later is more prevalent to Eastern European countries.

Some other issues:

- You should take the citation to the end of the sentence, as it is common in the literature.

- The methodology section should include all techniques, not only the OLS you use. Later you apply the t-test for post-estimation. This part should also be in the methodology. At the same time, in the result section there should be only your results without referencing to others. If you make comparison to other’s result then it is not result but analysis.

- The result section should be divided into result that include only the results and the accept/reject of the hypothesis. An analysis part should interpret the results in more details including to reference to other papers/results.

- the discussion part should be a summary

- there should be no further discussion about the control variables or you should make hypothesis about them.

6. PLOS authors have the option to publish the peer review history of their article (what does this mean?). If published, this will include your full peer review and any attached files.

Reviewer #1: No

Reviewer #2: No

---

## [Author Response · Author response to Decision Letter 0]

14 Jan 2021

RESPONSE TO REVIEWERS 

Dear reviewers please find our response to each of your point below . The structure is : your comment - our answer. 

1. Is the manuscript technically sound, and do the data support the conclusions?

Reviewer #1: Partly

Answer: Thank you for making this important point. We have fully taken your comments on board and completely revised the abstract, introductory part and the conceptual framework. We were able to go into more details of our finding in the analysis (section6) and summary (section7). We very much hope that they are now clearer and offer a clear indication as to the purpose of the paper.

The revised paper clarifies our main aim - to examine how various institutional arrangements (pillars) influence both the productive and unproductive entrepreneurial activity in entrepreneurial ecosystems in cities. 

Reviewer #2: Yes

Many thanks for your positive feedback. 

2. Has the statistical analysis been performed appropriately and rigorously? 

 Reviewer #1: Yes

Reviewer #2: Yes

Answer:

Many thanks for your positive feedback. 

3. Have the authors made all data underlying the findings in their manuscript fully available?

Reviewer #1: No

Reviewer #2: Yes

Answer: Apologies we were not aware of The PLOS Data policy that requires authors to make all data underlying the findings described in their manuscript fully available without restriction. This is now done with both dofile and datafile were uploaded as supplementary materials. For data description and sample representativeness please refer to section 4.1. 

4. Is the manuscript presented in an intelligible fashion and written in standard English?

Reviewer #1: Yes

Reviewer #2: Yes

Answer: Thank you.

5. Review Comments to the Author

Response to Reviewer #1: 

This manuscript has the potential to contribute to research on regional entrepreneurship and urban studies. The argument is generally clear and supported by data. The methods are robust. The paper is sufficiently grounded in relevant literature. My recommendations are as follows:

Major recommendation:

Further investigate the role of institutions at multiple scales. Much regional studies/urban studies/economic geography literature emphasizes the importance of analyzing not only national institutions, but also regional and local. Yet local and regional institutions are analyzed rather indirectly here. The authors consider the GEDI index (scaled at the national level) and a survey question about corruption (the question asks whether local 'political entrepreneurs' engage in corrupt practices with national governments). Since this paper is focused on urban entrepreneurship, it seems odd that the paper does not consider the effect of urban policy or urban institutions.

Answer: Thank you for pointing out this very important point. We have now articulated that this study contributes to the institutional and entrepreneurship ecosystems literature by introducing and examining a novel, multi-dimensional institutional context of cities in developing and transition economies, capturing variation that, we argue, affects both the productive and unproductive of entrepreneurial activity in a city. 

We dropped the GEDI index (scaled at the national level) and we draw from institutional theory (Scott, 1995; Busenitz et al., 2000) by creating the three dimensions of institutional arrangement – regulatory, cognitive, and normative institutional arrangements for cities. We also introduce an emerging role of civil society to facilitate productive and reduce unproductive entrepreneurship in ecosystems. We collected additional data, in particular on the number of business schools (cognitive pillar), environmental awareness and civil society – number of non-for -profits (normative pillar). Please refer to section 2.2. for the description and theorization of three pillars at city level. Our theory and analysis from now on are aligned at city level. 

Minor recommendations:

1) Clarify the research question. Perhaps I am misreading, but the research question does not seem relevant to the subsequent hypotheses ('what if the entrepreneurial activity emanating from an entrepreneurial ecosystem is not productive?' p2). The question is about negative factors that inhibit entrepreneurship (e.g. that paragraph references Baumol's 'destructive entrepreneurship' concept). But the hypotheses are about the dynamics of entrepreneurial ecosystems in developing countries (informal networks and institutional quality).

Answer: Thank you for drawing our attention to this. We apologise for misunderstanding the research question. We have thoroughly revised the introduction, theory and analysis and conclusions. This paper aim is to examine how various institutional arrangements (pillars) influence both the productive and unproductive entrepreneurial activity in entrepreneurial ecosystems in cities. To date, scholarly progress in this area has been limited mainly by a measurement challenge (Stenholm et al. 2013; Stam, 2018; Stam and van de Ven, 2019; Leendertse et al. 2020) —the measures in use fail to capture the multi-facet and heterogeneous nature of the institutional context phenomena for entrepreneurship. Our work contributes to the institutional and entrepreneurship ecosystems literature by introducing and examining a novel, multi-dimensional institutional context at city-level in developing and transition economies, capturing variation that, we argue, affects both the productive and unproductive of entrepreneurial activity in a city.

Following your valuable suggestion, we formulated and tested three research hypothesis (please refer to section 2.2.).

 In addition we added data on cities in Croatia and Bulgaria altering the number of cities to 16 and countries to 9. WE now use 1652 individual observations.

2) Provide more details on the survey sampling method. Why and how were the respondents selected? What makes them representative authorities on their respective cities? And why is the response rate so high (90%)?

Answer: Please refer to section 4.1. for the dull description of data collection process and method. We were able to continue data collection since we submitted the paper. In the original submission we had the 1278 individuals from Warsaw and Wroclaw in Poland; Lviv and Kyiv in Ukraine, Cluj and Bucharest in Romania, Astana and Almaty in Kazakhstan; Batumi and Tbilisi in Georgia; Istanbul in Turkey and Sarajevo in Bosnia and Herzegovina. We were able to increase the sample by adding two cities in Croatia (Zagreb and Osijek) and two cities in Bulgaria (Plovdiv and Sofia) bringing the total number of observations to 1,652. The data has been collected between December 2018 and January 2020. 

Dear reviewer 1. 

We are grateful to you for your time and constructive feedback. As was pointed out we have addressed these issues in the revised paper. In particular in the introduction, theory, method as well as the discussion of the results. We took a more detailed approach reporting results (section 5), discussion on finding with the analysis part (section 6) and summary of the paper (section 7). 

We have substantially revised the front part of the paper and added more observation for analysis addressing key issues you and the second reviewer raised as well as ensuring it reads more fluently. To this end we have the paper copy-edited professionally. 

We like to take this opportunity to thank you again for your effort to help us to make this work stronger. We have done our utmost to comply.

Response to Reviewer #2: 

Reviewer #2: Overall, the paper is dealing with an important issue the role of networking and connectivity in relation to institutional quality in EE. The main focus of the paper is EE in the developing countries. In these respects, the paper provides novel contribution to the EE literature. The authors use a unique dataset to test their hypothesis. While the paper deserves publishing in Plos One. However, I am suggesting some changes before publishing.

Answer. Thank you very much for your positive feedback. 

Reviewer #2: First, it seems that this paper is not the final version that the authors wanted to submit. There are some unfinished sentences and double words. In addition, while there are some references in the result section to hypothesis 3, but I cannot find hypothesis 3, only hypothesis 1 and 2.

Answer: Thank you for pointing out on this. It was a typo. However we now build stronger on institutional theory and develop 3 research hypotheses that test the relationship between three pillars (normative, cognitive, regulatory) of institutional arrangements in cities and the type of entrepreneurship (Baumol, 1990, 1993)- productive and unproductive. 

Reviewer 2: The concept section is the weakest part of the paper and should be rewritten by centering on the core concept. Right now, it is rather a reading list about the importance of networks and connectivity in the EE. It seems, that the authors follow the Stam (2015) model but it is not explicit. Later they refer to the systemic conditions in the Stam model as it is equal to the network. However, networking is only one element in the Stam concept. I think it needs more clarification. First, what are the elements of the EE and how networking is connected to these elements. The development issue should come after the core model is clear.

Answer: 

Thank you for your valuable and insightful points. Reflecting on the points you make we offered the core model. Drawing on the works of Scott (1995) and Stenholm et al. (2013) on the role of the institutional context for entrepreneurial activity and more recent works of Stam (2015, 2018), Audretsch and Belitski (2017) on the governance of entrepreneurial ecosystems that stimulate productive and high-growth entrepreneurship in regions and cities, we distinguished the definitive role of institutions in the outcomes of the entrepreneurial ecosystem. We therefore developed our model with this in mind. Accordingly, we changed the title of the paper to better reflect on the improvement of the theoretical model and what contribution it makes. 

Please refer to sections 2.1. and 2.2. for model design and hypothesis formulation. We hope that the storyline is now clear as is our rationale for focusing on the institutional pillars in cities to promote productive entrepreneurship and deter unproductive is now made clearer. 

Reviewer 2: Regarding the hypothesis I would add another hypothesis about the effect of culture also.

Answer: 

Please refer to section 2.2. with the hypothesis on culture is now part of the normative pillar of institutions (H3). 

Reviewer 2: Regarding Baumol you should refer not only to destructive but also to unproductive entrepreneurship, that later is more prevalent to Eastern European countries.

Answer:

Many thanks for drawing our attention to this. Indeed, this is a very valuable point which further pivoted our research (please refer to section 4.2.).

Our dependent variables now include measures of productive and unproductive entrepreneurial activity in the ecosystem. We study productive entrepreneurship activity (EE quality) with the following survey question "There is a strong focus on growth-oriented and productive entrepreneurship activity in my region (city)" measured on the Likert scale from 1 – very weak to 7 – very strong (Stam, 2018). This measure has been used in Baumol (1990) as well as Sobel (2008) as well as more recent study on the role of institutions for entrepreneurial quality cross-country (Chowdhury et al. 2019).

We use the question in a survey which measures the degree of unproductive entrepreneurship in a city based on the Sobel and Garrett (2002) and Chowdhury et al. (2019) studies. Unproductive entrepreneurship represents the unethical behavior of firms and necessity-driven entrepreneurship activity, which is particularly relevant for transition and developing countries (McMullen et al., 2008). We define Unproductive entrepreneurship with the following survey question "There is an economic activity of entrepreneurs via formal and informal cooperation with the local (national) government to access resources in a privileged way compared to other entrepreneurs whose access to resources could be limited or restricted measured on the Likert scale from 1 – very weak to 7 – very strong (Sobel, 2008). 

Reviewer 2: Some other issues:

- You should take the citation to the end of the sentence, as it is common in the literature.

Answer:

Done throughout the text where possible and feasible. 

Reviewer 2: The methodology section should include all techniques, not only the OLS you use. Later you apply the t-test for post-estimation. This part should also be in the methodology. 

Answer: Please refer to section 4.5 for the details of method description. the t-test for post-estimation is not used in our new model.

Reviewer 2: 

At the same time, in the result section there should be only your results without referencing to others. If you make comparison to other’s result then it is not result but analysis.

- The result section should be divided into result that include only the results and the accept/reject of the hypothesis. An analysis part should interpret the results in more details including to reference to other papers/results.

- the discussion part should be a summary

Answer: we followed your advise and we restructured the result section which is now includes section results (section 5) that includes only the results and the accept/reject. We added section 6 which includes further discussion and interprets the results in more details, compared the prior literature and empirical works. WE created a new section summary (section 7 – please refer for details). 

We are grateful to the reviewer for identifying this omission and apologies for our oversight. As was pointed out we have addressed these issues in the revised paper. In particular in the introductory, theory, method and discussion sections. We have substantially revised the paper addressing key issues you raise as well as ensuring it reads more fluently and is free from obvious typos and glitches. To this end we have the paper copy-edited professionally. 

---

## [Decision Letter · Decision Letter 1]

10 Feb 2021

Entrepreneurial Ecosystems in Cities: The Role of Institutions

PONE-D-20-27940R1

Dear Dr. Belitski,

We’re pleased to inform you that your manuscript has been judged scientifically suitable for publication and will be formally accepted for publication once it meets all outstanding technical requirements.

Kind regards,

Laurentiu Rozylowicz, Ph.D.

Academic Editor

PLOS ONE

Additional Editor Comments (optional):

Reviewers' comments:

Reviewer's Responses to Questions

**Comments to the Author**

1. If the authors have adequately addressed your comments raised in a previous round of review and you feel that this manuscript is now acceptable for publication, you may indicate that here to bypass the “Comments to the Author” section, enter your conflict of interest statement in the “Confidential to Editor” section, and submit your "Accept" recommendation.

Reviewer #1: (No Response)

Reviewer #2: All comments have been addressed

2. Is the manuscript technically sound, and do the data support the conclusions?

Reviewer #1: Yes

Reviewer #2: Yes

3. Has the statistical analysis been performed appropriately and rigorously? 

Reviewer #1: Yes

Reviewer #2: Yes

4. Have the authors made all data underlying the findings in their manuscript fully available?

Reviewer #1: Yes

Reviewer #2: Yes

5. Is the manuscript presented in an intelligible fashion and written in standard English?

Reviewer #1: Yes

Reviewer #2: Yes

6. Review Comments to the Author

Reviewer #1: The authors have effectively addressed my original recommendations. I think this manuscript will be ready for publication pending very minor edits for clarity. The manuscript needs a good proofreading for typos and grammar. And I encourage caution when arguing in the abstract that "this study is the first" of its kind. I'm not sure that's accurate (while each discipline has it's own methodologies, this kind of quantitative urban institutional research is relatively common in economic geography and regional science journals).

Reviewer #2: The paper has improved a lot as compared to the previous version. I am satisfied with the changes and have no further suggestions.

7. PLOS authors have the option to publish the peer review history of their article (what does this mean?). If published, this will include your full peer review and any attached files.

Reviewer #1: No

Reviewer #2: No

---

## [Editor Report · Acceptance letter]

11 Feb 2021

PONE-D-20-27940R1 

Entrepreneurial Ecosystems in Cities: The Role of Institutions 

Dear Dr. Belitski:

I'm pleased to inform you that your manuscript has been deemed suitable for publication in PLOS ONE. Congratulations! Your manuscript is now with our production department. 

Kind regards, 

on behalf of

Dr. Laurentiu Rozylowicz 

Academic Editor

PLOS ONE